# Replication Study: The microRNA miR-34a inhibits prostate cancer stem cells and metastasis by directly repressing CD44

**Xuefei Yan, Beibei Tang, Biao Chen, Yongli Shan, Huajun Yang, Reproducibility Project: Cancer Biology***

Crown Biosciences Inc, Science & Technology Innovation Park, Taicang, China

**Abstract** As part of the Reproducibility Project: Cancer Biology, we published a Registered Report (Li et al., 2015), that described how we intended to replicate selected experiments from the paper 'The microRNA miR-34a inhibits prostate cancer stem cells and metastasis by directly repressing CD44' (Liu et al., 2011). Here we report the results. We found the microRNA, miR-34a, was expressed at twice the level in CD44$^+$ prostate cancer cells purified from xenograft tumors (LAPC4 cells) compared to CD44$^-$ LAPC4 cells, whereas the original study reported miR-34a was underexpressed in CD44$^+$ LAPC4 cells (Figure 1B; Liu et al., 2011). When LAPC4 cells engineered to express miR-34a were injected into mice, we did not observe changes in tumor growth or CD44 expression; however, unexpectedly miR-34a expression was lost *in vivo*. In the original study, LAPC4 cells expressing miR-34a had a statistically significant reduction in tumor regeneration and reduced CD44 expression compared to control (Figure 4A and Supplemental Figures 4A,B and 5C; Liu et al., 2011). Furthermore, when we tested if miR-34a regulated CD44 through binding sites in the 3'UTR we did not find a statistically significant difference, whereas the original study reported miR-34a decreased CD44 expression that was partially abrogated by mutation of the binding sites in the *CD44* 3'UTR (Figure 4D; Liu et al., 2011). Finally, where possible, we report meta-analyses for each result.

DOI: https://doi.org/10.7554/eLife.43511.001

*For correspondence:
tim@cos.io;
nicole@scienceexchange.com

Group author details:
Reproducibility Project: Cancer
Biology See page 15

Competing interest: See
page 15

## Introduction

The Reproducibility Project: Cancer Biology (RP:CB) is a collaboration between the Center for Open Science and Science Exchange that seeks to address concerns about reproducibility in scientific research by conducting replications of selected experiments from a number of high-profile papers in the field of cancer biology (Errington et al., 2014). For each of these papers a Registered Report detailing the proposed experimental designs and protocols for the replications was peer reviewed and published prior to data collection. The present paper is a Replication Study that reports the results of the replication experiments detailed in the Registered Report (Li et al., 2015) for a paper by Liu et al., and uses a number of approaches to compare the outcomes of the original experiments and the replications.

Many cancers contain a subset of cells, termed cancer stem cells (CSCs), that are less differentiated and contribute to tumor progression and metastasis in various cancers (Kim et al., 2018; Takayama et al., 2017). In 2011, Liu et al., reported that the microRNA miR-34a suppressed prostate CSCs via the adhesion molecule CD44 (Liu et al., 2011). miR-34a was reported to be repressed in CD44$^+$ prostate CSCs, below levels of let-7b, a previously reported tumor suppressive microRNA (Liu et al., 2012; Liu et al., 2011). Enforced expression of miR-34a, such as through lentiviral infection of the prostate cancer cell line LAPC4, inhibited tumor regeneration (Liu et al., 2011). CD44

was also reported to be a direct and functional target of miR-34a, suggesting a key role of miR-34a in regulating prostate CSCs (*Liu et al., 2011*).

The Registered Report for the paper by Liu et al. described the experiments to be replicated (Figures 1B, 4A and D and Supplemental Figures 4A,B and 5C), and summarized the current evidence for these findings (*Li et al., 2015*). Since that publication additional studies have reported tumor suppressive functions of miR-34a in other types of cancers. This includes inhibition of cellular proliferation and invasion in endometrial cancer by miR-34a through downregulating Notch1 (*Wang et al., 2017*), inhibition of cellular migration and invasion in hepatocellular carcinoma via sirtuin 1 (*Zhou et al., 2017*), inhibition of cellular proliferation and clonogenicity of breast CSCs by directly preventing RNA 2',3'-Cyclic Phosphate and 5'-OH Ligase accumulation (*Lin et al., 2017*), inhibition of proliferation and migration of gastric cancer cells through CD44 suppression (*Jang et al., 2016*), and inhibition of tumor growth in non-small cell lung cancer with a combination of miR-34a and let-7 (*Kasinski et al., 2015*). In addition to miR-34a, miR-141 was identified to suppress prostate CSCs by targeting a cohort of pro-metastasis genes including *CD44* (*Liu et al., 2017*). Similar to miR-34a, miR-141 was underexpressed in CD44$^+$ prostate CSCs, inhibited tumor regeneration when expressed in prostate cancer cell lines, and regulated *CD44* through a putative miR-141 binding site in the 3'UTR of *CD44* (*Liu et al., 2017*). In addition, miR-34a delivered via a nanoparticle (MRX34)

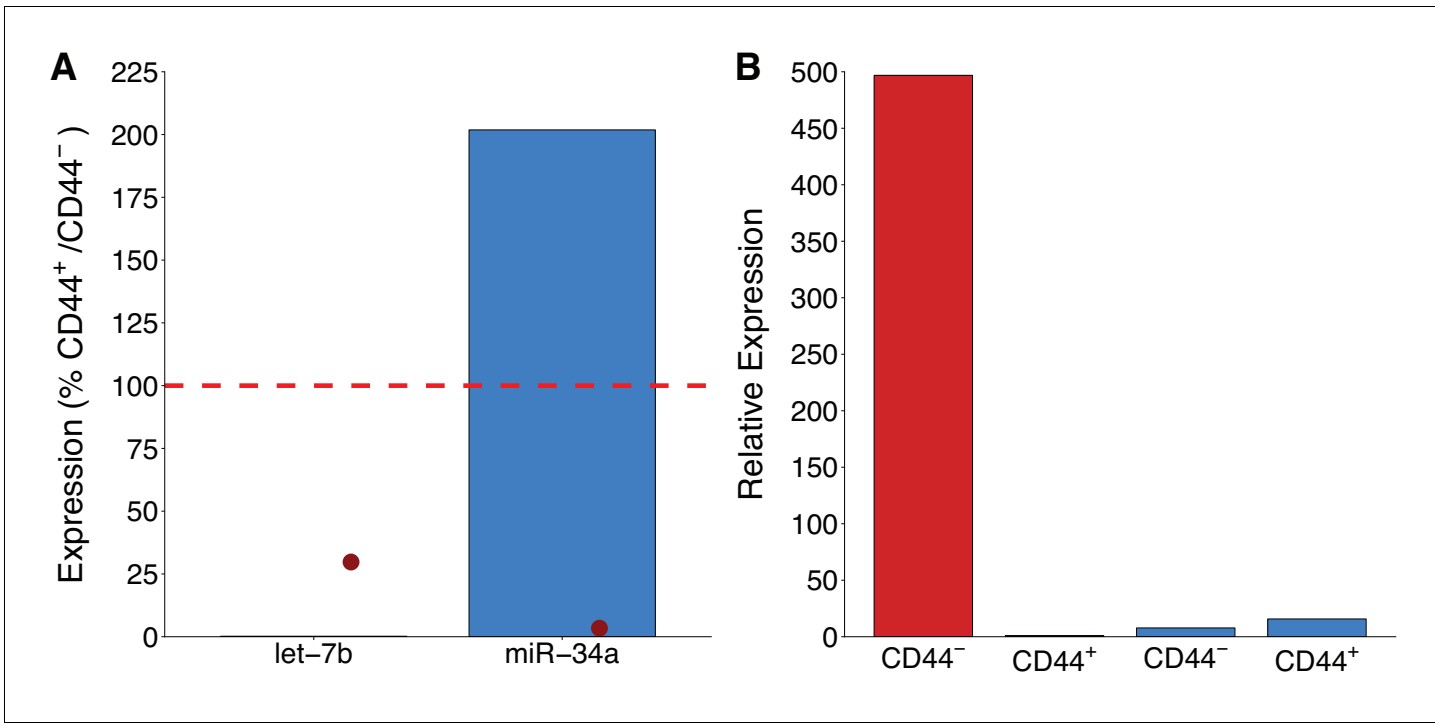

**Figure 1.** miR-34a expression in CD44 populations. LAPC4 cells purified from xenograft tumors were sorted into CD44$^+$ and CD44$^-$ populations by fluorescence-activated cell sorting. (A) Percent expression of let-7b or miR-34a (normalized to miR-103) was determined by qRT-PCR for both populations. For each microRNA, expression levels in CD44$^+$ cells were made relative to CD44$^-$ cells, such that if expression was equal it would be assigned a value of 100% (indicated by dashed line). Means reported from one biological repeat. Note, let-7b expressed at 0.20%. Data from the representative experiment reported in Figure 1B of *Liu et al. (2011)* is displayed as a single point (red circle) for comparison. (B) Relative expression levels of let-7b or miR-34a (normalized to miR-103) are presented for CD44$^-$ and CD44$^+$ populations. Expression levels were made relative to let-7b in CD44$^+$ cells which were assigned a value of 1. This is the same experiment as part (A), which is from one biological repeat. Additional details for this experiment can be found at https://osf.io/tkn6c/.

DOI: https://doi.org/10.7554/eLife.43511.002

The following figure supplement is available for figure 1:

**Figure supplement 1.** Flow cytometry gating strategy and controls.
DOI: https://doi.org/10.7554/eLife.43511.003

has been tested in a clinical trial to treat patients with unresectable primary hepatocellular carcinoma or individuals with unresectable liver metastasis (*Bouchie, 2013*); however, the trial was terminated after five immune related serious adverse events (ClinicalTrials.gov identifier: NCT01829971). There have also been numerous studies reporting the utility of miR-34a as a biomarker, along with other biomarkers, to increase the sensitivity of detection in breast cancer (*Zaleski et al., 2018*) and oral cancer (*Shah et al., 2018*).

The outcome measures reported in this Replication Study will be aggregated with those from the other Replication Studies to create a dataset that will be examined to provide evidence about reproducibility of cancer biology research, and to identify factors that influence reproducibility more generally.

## Results and discussion

For this study, we obtained a sample of androgen dependent LAPC4 cells from the authors of the original study. Interestingly, while the short tandem repeat (STR) profile of these cells was a partial match with the LAPC4 profiles in the DSMZ and Cellosaurus (*Bairoch, 2018*) databases, it was not a match for the amelogenin locus, which is used in sex determination (*Table 1*). Instead of containing the X and Y alleles as expected, these cells only contained the X allele. This could have been due to random amplification failure (i.e. allelic dropout) or deletion of the Y allele (*Ou et al., 2012*; *Xu et al., 2017*). However, these cells also did not form tumors when injected into male NOD/SCID mice as described in Protocol 1 of the Registered Report. We obtained another sample of cells, but this time from the authors who originally isolated the LAPC4 cell line (*Klein et al., 1997*). These cells had a partial match with the database profiles for LAPC4 cells, including the X and Y alleles for the amelogenin locus (*Table 1*), and, importantly, formed tumors when implanted into mice. Thus, we used these LAPC4 cells for the experiments described below.

### Expression of miR-34a in LAPC4 cells

We independently replicated an experiment to determine miR-34a expression in CD44$^+$ and CD44$^-$ LAPC4 cells. This experiment is comparable to what was reported in Figure 1B of *Liu et al. (2011)* and described in Protocol 3 in the Registered Report (*Li et al., 2015*). While the original study included other prostate cancer cell lines, this replication attempt was restricted to LAPC4 cells. LAPC4 cells purified from xenograft tumors were sorted into CD44$^+$ and CD44$^-$ populations by fluorescence-activated cell sorting (FACS). However, while the original study, and the described protocol in the Registered Report, described the top 10% most brightly stained cells to be defined as CD44$^+$ and the bottom 10% most dimly stained cells as CD44$^-$, we observed that the CD44 signal was weak without a clear division between positive and negative populations (*Figure 1—figure supplement 1A*). Additionally, the CD44 stained cell population was largely indistinguishable from cells stained with an isotype control antibody. Importantly, the isotype control and CD44 antibodies were matched on heavy chain, subclass, light chain class, fluorochrome type, fluorochrome ratio, manufacturing process, and formulation, which are known factors that impact the usability of the isotype control in flow cytometry (*Hulspas et al., 2009*; *Maecker and Trotter, 2006*). We also confirmed the CD44 antibody was able to successfully detect CD44$^+$ human peripheral blood mononuclear cells (*Figure 1—figure supplement 1B*). Considering the weakly positive sample, CD44$^+$ cells were defined as the top 1% of the population with the remaining population constituting CD44$^-$ cells, a deviation from the original study and the Registered Report. We then used quantitative real-time-polymerase chain reaction (qRT-PCR) to assess let-7b and miR-34a expression in CD44$^+$ and CD44$^-$ LAPC4 cell populations. We found that let-7b was underexpressed in CD44$^+$ cells, being expressed at 0.20% of the level in the CD44$^-$ cells, while miR-34a was elevated in level in CD44$^+$ cells, being expressed at 202% of the level in the CD44$^-$ cells (*Figure 1*). The original study reported both let-7b and miR-34a were underexpressed in CD44$^+$ LAPC4 cells at 30% and 3.4% of the level in CD44$^-$ cells, respectively (*Liu et al., 2011*). The decision to define the CD44$^+$ population as a smaller fraction of the total population than the original study (1% vs 10%) was done to minimize any potential masking of reduced microRNA expression by not including what would have been a relatively large population (90%) of cells that were not positive for CD44 based on controls. Importantly, we observed let-7b levels were reduced in CD44$^+$ cells, relative to CD44$^-$ cells, as expected and similar to the original study despite the difference in how populations were defined.

However, this creates a potential confound, particularly in the CD44⁻ population as defined for this replication, that could contain cells that were positive for CD44 complicating the comparison of microRNA levels between these populations. To summarize, for this experiment we found results that were in the same direction as the original study for let-7b expression, but not for miR-34a expression.

## Effect of miR-34a expression on tumor growth

LAPC4 tumor cells were used to replicate an experiment that tested the effect of miR-34a on tumor growth. This experiment is similar to what was reported in Supplemental Figure 5C of *Liu et al. (2011)* and described in Protocol 4 in the Registered Report (*Li et al., 2015*). Untransduced LAPC4 cells, which were not included in the original study, were added as an additional control. LAPC4 cells purified from xenograft tumors were transduced with the same lentiviral vectors used in the original study to create stable miR-34a producing cells, which expressed higher levels of miR-34a compared to untransduced or negative control transduced LAPC4 cells (*Figure 2A*). Following infection, cells were injected into male NOD/SCID mice and monitored for tumor development. Tumor incidence was 100% for all groups, similar to the original study, with the three groups having the same median latency time (*Figure 2—figure supplement 1A*). After 100 days, tumors were excised and weighed. Tumors from negative control transduced cells grew to an average of 0.51 grams [n = 7, *SD* = 0.28] (*Figure 2B*) similar to the weight reported for this group in the original study [0.54 grams, *SD* = 0.32, n = 6], which was the planned stopping rule in the Registered Report. The original study observed this at 69 days post injection. The original study reported expression of miR-34a inhibited tumor regeneration of LAPC4 cells [0.15 grams, *SD* = 0.09, n = 6] (*Liu et al., 2011*), while in this replication attempt, we observed LAPC4 cells engineered to express miR-34a resulted in similar tumor weights [0.55 grams, *SD* = 0.17, n = 7] as the untransduced or negative control transduced LAPC4 cells (*Figure 2B*). The planned comparisons of the miR-34a group to the negative control group (t(18) = 0.432, *p*=0.671) or the untransduced group (t(18) = 0.414, *p*=0.684) were not statistically significant. In addition to collecting tumor weights, we monitored tumor volume, determined by caliper, during the course of the study and observed similar growth profiles for the three groups (*Figure 2C*, *Figure 2—figure supplement 1B*). To summarize, we found results that were not in the same direction as the original study and not statistically significant.

The excised tumors were further assessed for miR-34a and CD44 expression. We measured miR-34a expression by qRT-PCR. This experiment was described in Protocol 6 in the Registered Report (*Li et al., 2015*) and is conceptually similar to what was reported in Supplemental Figure 4A,B of *Liu et al. (2011)*. The original study reported no, or marginal (~3X), increases in miR-34a expression in the residual tumors formed from prostate cancer cells, specifically LAPC4, LAPC9, and DU145, transiently transfected with miR-34a oligo compared to vector control (*Liu et al., 2011*). An increase (~3X over control) in miR-34a expression was also reported for tumors derived from DU145 cells transduced with lentiviral vectors to express miR-34a (*Liu et al., 2011*). In this replication attempt, we detected miR-34a expression in tumors derived from untransduced or negative control transduced LAPC4 cells; however, unexpectedly we were unable to detect miR-34a in the tumors derived from miR-34a transduced LAPC4 cells (*Figure 2D*). Although miR-34a was not detectable in the tumors derived from miR-34a transduced LAPC4 cells, we were able to detect the copepod green fluorescent protein (copGFP) marker, which was under control of the elongation factor-1 (EF-1) alpha promoter. This suggests that the pre-miR-34a that successfully expressed mature miR-34a in LAPC4 cells *in vitro* (*Figure 2A*) was silenced *in vivo* (*Figure 2D*). This could have occurred through promoter hypermethylation or chromatin remodeling of the locus where the synthetic construct was expressed. Indeed, tumor suppressor microRNAs have been reported to be downregulated, or completely deleted, in a variety of cancers (*Kelly and Russell, 2009*; *Rusek et al., 2015*). Additionally, the cytomegalovirus (CMV) promoter, which controlled pre-miR-34a expression, has been reported to become silent in many animal studies within a few weeks after gene transfer (*Löser et al., 1998*), which could be another mechanism responsible for this phenotype. Importantly, though, this observation does not necessarily imply miR-34a does not suppress tumor growth or progression in this experimental model since miR-34a overexpression was not confirmed *in vivo*. The different source of the LAPC4 cells between the original study and this replication attempt might have contributed to differences in miR-34a expression. It is also possible that silencing of miR-34a could

have imparted a benefit at an early initiation step after cells were implanted in the mice and been maintained during tumor growth.

We also measured CD44 expression by Western blot in the excised tumors. This experiment was described in Protocol 5 in the Registered Report (*Li et al., 2015*) and is conceptually similar to what was reported in Figure 4A of *Liu et al. (2011)*, which used DU145 cells transduced with a retroviral vector encoding miR-34a. The original study reported CD44 expression was inversely correlated with miR-34a expression in the tumors formed from the transduced DU145 cells (*Liu et al., 2011*). In this replication attempt, we found CD44 expression in tumors derived from miR-34a transduced LAPC4 cells was similar to the expression in tumors derived from untransduced or negative control transduced LAPC4 cells (*Figure 2E*, *Figure 2—figure supplement 1D*). Interpretation of these results need to take into consideration the absence of miR-34a expression that was observed in the tumors from miR-34a transduced LAPC4 cells.

## Evaluation of putative miR-34a binding sites in the 3'UTR of CD44

We independently replicated an experiment to test if miR-34a regulates *CD44* expression through two putative miR-34a binding sites in the 3'UTR of *CD44*. DU145 cells were co-transfected with miR-34a oligos, or control oligos, and the same luciferase reporter plasmid used in the original study that contained a fragment of the 3'UTR of *CD44* with both putative miR-34a binding sites that were either left intact (wild-type) or mutated. This is similar to what was reported in Figure 4D of *Liu et al. (2011)* and described in Protocol 7 in the Registered Report (*Li et al., 2015*). While the original study tested each binding site individually as well as together as a double mutant, this replication attempt focused on the double mutant. We found DU145 cells transfected with the luciferase reporter, with wild-type binding sites, and miR-34a oligos were similar (mean relative expression = 0.99) to cells co-transfected with control oligos (mean relative expression = 1.00), which was not statistically significant (one-sample Wilcoxon signed-rank test: $z = 0.517$, uncorrected $p=0.632$; (Bonferroni corrected $p>0.99$) (*Figure 3A*). Results were similar when DU145 cells were transfected with the luciferase reporter containing mutations in both binding sites (miR-34a: mean relative expression = 1.19; control: mean relative expression = 1.00). The original study reported cells co-transfected with the wild-type luciferase reporter and miR-34a oligos resulted in a decreased luciferase signal (mean relative expression = 0.67) compared to cells co-transfected with control oligos (mean relative expression = 1.00), which was partially abrogated with the double mutant luciferase reporter (miR-34a: mean relative expression = 0.78; control: mean relative expression = 1.00) (*Liu et al., 2011*). In addition to analyzing the data like the original study, where luciferase values from cells transfected with miR-34a oligos were reported normalized to control oligo values for each biological repeat and each luciferase reporter, we also analyzed the data using the raw luciferase values. While we did not observe a difference between miR-34a and control oligos for each luciferase reporter, in agreement with the results of the prior analysis approach, we found the luciferase values from the double mutant luciferase reporter was on average lower than the wild-type luciferase reporter values (*Figure 3B*). Importantly, interpretation of results from experiments that overexpress microRNAs should take into account artificial repression that could occur from non-physiological stoichiometry of the overexpressed microRNA as well as relieved repression of natural targets of other microRNAs (*Svoboda, 2015*). To summarize, we found results that were not consistent in direction with the original study and not statistically significant.

## Meta-analysis of original and replication effects

We performed a meta-analysis using a random-effects model, where possible, to combine each of the effects described above as pre-specified in the confirmatory analysis plan (*Li et al., 2015*). To provide a standardized measure of the effect, a common effect size was calculated for each effect from the original and replication studies. Cohen's *d* is the standardized difference between two independent means using the pooled sample standard deviation. For a one-sample test, Cohen's *d* is the difference between the sample mean and the null value using the sample standard deviation. The effect size Glass' delta is a standardized difference between two means using the standard deviation of only the control group. Glass' delta was used when the variance between the two conditions were not equal, which occurred in the original study for some of the experiments. The estimate of the effect size of one study, as well as the associated uncertainty (i.e. confidence interval (CI)),

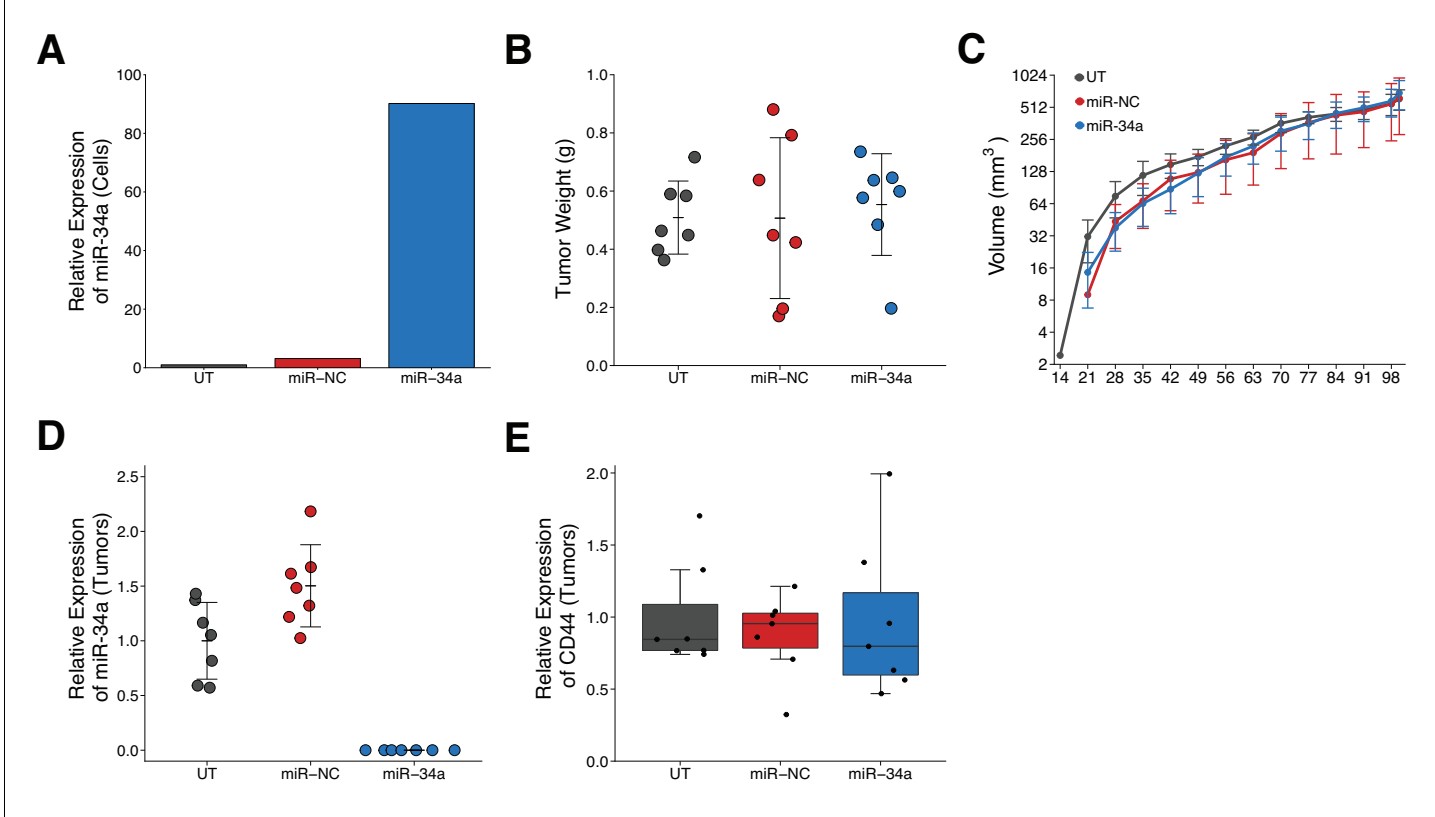

**Figure 2.** Effect of miR-34a expression on tumor growth and CD44 expression in LAPC4 tumors. Purified LAPC4 cells were transduced to express miR-34a before injection into male NOD/SCID mice. (A) Relative expression levels of miR-34a (normalized to miR-103) was determined by qRT-PCR for LAPC4 cells left untransduced (UT), or transduced with lentivirus encoding scramble negative control (miR-NC) or pre-miR-34 (miR-34a). For each condition, expression level was made relative to UT which were assigned a value of 1. Means reported from one biological repeat. (B) At the end of the experiment (100 days post injection), primary tumors were excised and weighed. Dot plot with means reported as crossbars and error bars represent *SD*. Number of primary tumors per group (n = 7). One-way ANOVA on all three groups: $F(2,18) = 0.119$, *p*=0.888. Planned contrast between miR-34a and miR-NC: Fisher's LSD test, $t(18) = 0.432$, *p*=0.671. Planned contrast between miR-34a and UT: Fisher's LSD test, $t(18) = 0.414$, p=0.684. (C) Following primary tumor detection, caliper measurements were taken three times a week and used to calculate tumor volume. Line graph of tumor volume (y-axis is natural log scale) with means reported and error bars representing *SD*. Number of mice monitored per group (n = 7). One-way ANOVA on area under the curve (natural log-transformed) for all three groups: $F(2,18) = 0.916$, *p*=0.418. Planned contrast between miR-34a and miR-NC: Fisher's LSD test, $t(18) = 0.680$, *p*=0.505, Cohen's $d = -0.36$, 95% CI [−1.41, 0.70]. Planned contrast between miR-34a and UT: Fisher's LSD test, $t(18) = 0.673$, *p*=0.509, Cohen's $d = 0.36$, 95% CI [−0.70, 1.41]. (D) Relative expression levels of miR-34a (normalized to miR-103) was determined by qRT-PCR from excised tumors. For each condition, expression level was made relative to UT. Dot plot with means reported as crossbars and error bars represent *SD*. Of note, no signal was detected in any of the tumors derived from miR-34a transduced LAPC4 cells. (E) Relative expression levels of CD44 (normalized to ß-actin) was determined by Western blot from excised tumors. For each condition, expression level are presented relative to UT. Box and whisker plot with median represented as the line through the box and whiskers representing values within 1.5 IQR of the first and third quartile. Individual data points represented as dots. Additional details for this experiment can be found at https://osf.io/n9vrz/.
DOI: https://doi.org/10.7554/eLife.43511.005

The following figure supplement is available for figure 2:

**Figure supplement 1.** Latency and individual tumor xenografts.
DOI: https://doi.org/10.7554/eLife.43511.006

compared to the effect size of the other study provides another approach to compare the original and replication results (*Errington et al., 2014*; *Valentine et al., 2011*). Importantly, the width of the confidence interval for each study is a reflection of not only the confidence level (e.g. 95%), but also variability of the sample (e.g. *SD*) and sample size.

The comparison of tumor weights between tumors derived from miR-34a transduced LAPC4 cells compared to negative control, which were reported in *Figure 2B* of this study and Supplemental Figure 5C of *Liu et al. (2011)*, resulted in effect sizes that were in opposite direction between the

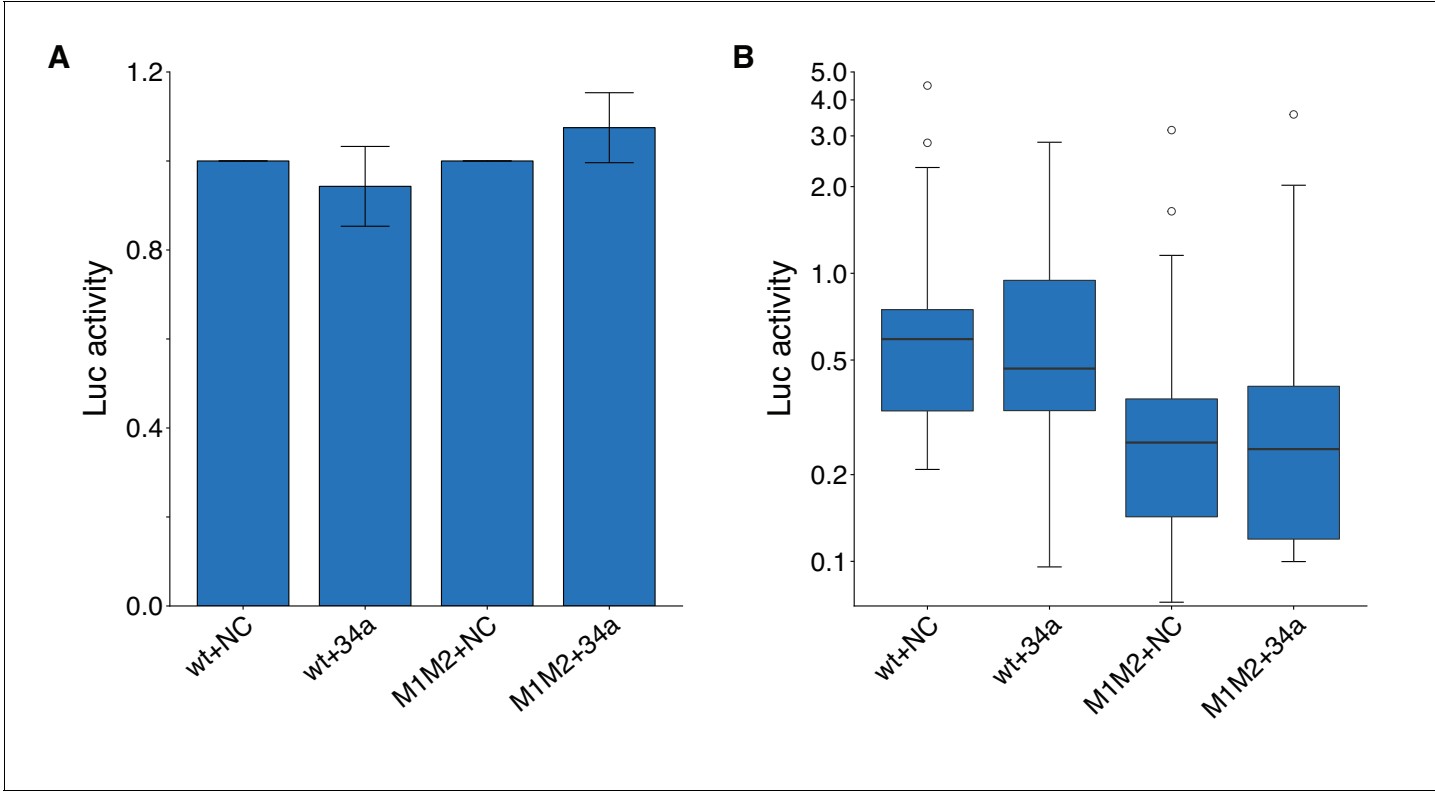

**Figure 3.** Luciferase assays to assess putative miR-34a binding sites in the 3'UTR of CD44. DU145 cells were transfected with either a luciferase reporter with a fragment of the 3'UTR of *CD44* with both putative miR-34a binding sites left intact (wt) or mutated (M1M2). Cells were also co-transfected with miR-34a oligos (34a) or control oligos (NC). (A) Luciferase (Luc) activity was normalized to the NC condition for both luciferase reporters for each biological repeat such that NC conditions were assigned a value of 1. Means reported and error bars represent s.e.m. from 16 independent biological repeats. Wilcoxon-Mann-Whitney test comparing Luc values from wt +34 a vs M1M2 + 34 a; $U$ = 104, uncorrected $p$=0.381, Bonferroni adjusted significance threshold = 0.0167; (Bonferroni corrected $p$>0.99). One-sample Wilcoxon signed-rank test comparing Luc values from wt +34 a to a constant of 1 (wt +NC); $z$ = 0.517, uncorrected $p$=0.632; (Bonferroni corrected $p$>0.99). One-sample Wilcoxon signed-rank comparing Luc values from M1M2 + 34 a to a constant of 1 (M1M2 + NC); $z$ = 0.310, uncorrected $p$=0.782; (Bonferroni corrected $p$>0.99). (B) This is the same data as in part (A), but Luc activity was not normalized. Luc values are presented relative to wt +NC. Box and whisker plot (y-axis is $\log_{10}$ scale) with median represented as the line through the box and whiskers representing values within 1.5 IQR of the first and third quartile. Kruskal-Wallis test on all four groups: H(3) = 11.1, $p$=0.011. Wilcoxon-Mann-Whitney comparison between luciferase reporter (wt or M1M2): $U$ = 266, uncorrected $p$=7.67×10$^{-4}$, Bonferroni corrected $p$=1.53×10$^{-3}$, Cliff's $d$ = 0.48, 95% CI [0.19, 0.69]. Wilcoxon-Mann-Whitney comparison between oligo (NC or 34a): $U$ = 497, uncorrected $p$=0.847, Bonferroni corrected $p$>0.99, Cliff's $d$ = 0.03, 95% CI [−0.25, 0.30]. Wilcoxon-Mann-Whitney comparison between wt +NC and wt +M1M2: $U$ = 119, uncorrected $p$=0.752, Bonferroni corrected $p$>0.99, Cliff's $d$ = 0.07, 95% CI [−0.32, 0.44]. Additional details for this experiment can be found at https://osf.io/vzb82/.

DOI: https://doi.org/10.7554/eLife.43511.007

two studies and the effect size point estimate of each study was not within the 95% CI of the other study (*Figure 4A*). The meta-analysis of these two effects was not statistically significant (*p*=0.510).

There were three comparisons made with the luciferase reporter data, which were reported in *Figure 3A* of this study and Figure 4D of *Liu et al. (2011)*. A comparison of luciferase values from cells transfected with miR-34a oligos, normalized to control oligo values, was made between wild-type and double mutant luciferase reporters. The effect sizes from the original study and this replication attempt were in the same direction and the effect size point estimates of each study were within the 95% CI of the other study (*Figure 4B*). The 95% CI for both studies also crossed zero, which indicates the null hypothesis that there is no difference between the two luciferase reporters can not be rejected. This is in agreement with the meta-analysis that was not statistically significant (*p*=0.353). Additionally, the luciferase values from miR-34a oligos normalized to control oligo values were compared to a constant of 1, which represents the control oligo group for each of the luciferase reporters (*Figure 4C*). This is analogous to a one-sample *t* test. For the wild-type luciferase reporter, the

original study and this replication attempt were in the same direction with the effect size of the replication within the 95% CI of the original study, but not vice versa. For the double mutant luciferase reporter, the effect sizes were in opposite directions between the two studies and the effect size point estimate of each study was not within the 95% CI of the other study. For both reporters, the 95% CI for both studies also crossed zero. We used the one-sample approach because of the way the original data were handled, where the control oligo for each biological repeat and each luciferase reporter were assigned values of 1 without an associated variance making an unpaired two-sample test inappropriate. This possibly explains why the original study reported a statistically significant effect between control and miR-34a for the wild-type luciferase reporter when using an unpaired test in contrast to the calculated 95% CI of the effect size for the original study presented in *Figure 4C* that crosses zero, which indicates the null hypothesis that there is no difference between miR-34a oligo transfected cells and the constant of 1 (i.e. control oligo) can not be rejected. This is in agreement with the meta-analyses that were not statistically significant (wild-type: $p=0.416$; double mutant: $p=0.581$). Additionally for these two comparisons, the large CI of the meta-analyses along with statistically significant Cochran's $Q$ tests (wild-type: $p=0.047$; double mutant: $p=0.021$) suggest heterogeneity between the original and replication studies.

This direct replication provides an opportunity to understand the present evidence of these effects. Any known differences, including reagents and protocol differences, were identified prior to conducting the experimental work and described in the Registered Report (*Li et al., 2015*). However, this is limited to what was obtainable from the original paper and through communication with the original authors, which means there might be particular features of the original experimental protocol that could be critical, but unidentified. So while some aspects, such as cell line, mouse strain, antibodies, and plasmids were maintained others were changed in the execution of the replication, such as source of the LAPC4 cells and flow cytometry gating strategy. Furthermore, other aspects were unknown or not easily controlled for. These include variables such as cell line genetic drift (*Ben-David et al., 2018*; *Hughes et al., 2007*; *Kleensang et al., 2016*), subclonal drift in heterogeneous stable cells (*Shearer and Saunders, 2015*), genetic heterogeneity of mouse inbred strains (*Casellas, 2011*), and the microbiome of recipient mice (*Macpherson and McCoy, 2015*). Whether these or other factors influence the outcomes of this study is open to hypothesizing and further investigation, which is facilitated by direct replications and transparent reporting.

## Materials and methods

**Key resources table**

| Reagent type (species) or resource | Designation | Source or reference | Identifiers | Additional information |
|---|---|---|---|---|
| Strain, strain background (*Mus musculus*, NOD/SCID, male) | NOD/SCID | Vital River | Strain code: 406; RRID:IMSR_CRL:394 | |
| Cell line (*Homo sapiens*, male) | LAPC4 | DOI: 10.1038/nm0497-402 | RRID:CVCL_4744 | shared by Dr. Dean G Tang, MD Anderson Cancer Center and Dr. Charles L Sawyers, Memorial Sloan Kettering Cancer Center |
| Cell line (*H. sapiens*, male) | DU145 | ATCC | cat#:HTB-81; RRID:CVCL_0105 | |
| Cell line (*H. sapiens*, male) | HT-1080 | ATCC | cat#:CCL-121; RRID:CVCL_0317 | |
| Cell line (*H. sapiens*) | HEK293T | ATCC | cat#:CRL-3216; RRID:CVCL_0063 | |
| Chemical compound, drug | testosterone propionate | Aladdin | CAS:57-85-2 | lot# L1401028 |

*Continued on next page*

*Continued*

| Reagent type (species) or resource | Designation | Source or reference | Identifiers | Additional information |
|---|---|---|---|---|
| Other | Matrigel | BD Biosciences | cat#:354248 | |
| Antibody | FITC-conjugated monoclonal anti-CD44 | BD Biosciences | cat#:555478; clone:G44-26; RRID:AB_395870 | 1:13 dilution |
| Antibody | FITC-conjugated monoclonal mouse IgG2b,κ isotype control | BD Biosciences | cat#:555742; clone:27–35; RRID:AB_396085 | 1:13 dilution |
| Antibody | rabbit anti-CD44 | AbCam | cat#:ab51037; clone:EPR1013Y; RRID:AB_868936 | 1:1000 dilution |
| Antibody | mouse anti-ß-actin | Cell Signaling Technology | cat#:3700; clone:8H10D10; RRID:AB_2242334 | 1:1000 dilution |
| Antibody | HRP-conjugated goat anti-rabbit | Cell Signaling Technology | cat#:7074; RRID:AB_2099233 | 1:2000 dilution |
| Antibody | HRP-conjugated horse anti-mouse | Cell Signaling Technology | cat#:7076; RRID:AB_330924 | 1:5000 dilution |
| Commercial assay, kit | MACS Lineage Cell Depletion Kit | Miltenyi Biotech | cat#:130-090-858 | |
| Recombinant DNA reagent | psPAX2 | Crown Biosciences, Inc | RRID:Addgene_12260 | |
| Recombinant DNA reagent | pMD2.G | Crown Biosciences, Inc | RRID:Addgene_12259 | |
| Recombinant DNA reagent | pMDLg/pRRE | Wuhan Miaoling Bioscience and Technology Co. | cat# P0685; RRID:Addgene_12251 | |
| Recombinant DNA reagent | pre-mir-34a | System Biosciences | cat# PMIRH34aPA-1 | |
| Recombinant DNA reagent | scramble non-targeting pre-miRNA | System Biosciences | cat# PMIRH000PA-1 | |
| Recombinant DNA reagent | pMIR-CD44-3'UTR | DOI: 10.1038/nm.2284 | | shared by Dr. Dean G Tang, MD Anderson Cancer Center |
| Recombinant DNA reagent | pMIR-CD44-M1M2-3'UTR | DOI: 10.1038/nm.2284 | | shared by Dr. Dean G Tang, MD Anderson Cancer Center |
| Recombinant DNA reagent | phRL-CMV | DOI: 10.1038/nm.2284 | | shared by Dr. Dean G Tang, MD Anderson Cancer Center |
| Sequence-based reagent | copGFP | DOI: 10.1111/j.1745-7270.2008.00448.x | | Forward: 5'-AGGACAGCGT GATCTTCACC-3'; Reverse: 5'-CTTGAAGTGCA TGTGGCTGT-3' |
| Sequence-based reagent | miR-34a miRNA TaqMan assay kit | Applied Biosystems | assayID:000426 | |
| Sequence-based reagent | has-let-7b miRNA TaqMan assay kit | Applied Biosystems | assayID:000378 | |
| Sequence-based reagent | miR-103 miRNA TaqMan assay kit | Applied Biosystems | assayID:000439 | |
| Sequence-based reagent | has-miR-34a-5p | Thermo Fisher Scientific | assayID:MC11030 | |

*Continued on next page*

*Continued*

| Reagent type (species) or resource | Designation | Source or reference | Identifiers | Additional information |
|---|---|---|---|---|
| Sequence-based reagent | negative control #1 | Thermo Fisher Scientific | cat#:4464058 | |
| Software, algorithm | GeneMapper | Applied Biosystems | RRID:SCR_014290 | version 4.0 |
| Software, algorithm | Studylog Systems | Studylog | RRID:SCR_016682 | version 3.1.399.19 |
| Software, algorithm | FACSDiva | BD Biosciences | RRID:SCR_001456 | version 6.1.3 |
| Software, algorithm | MxPro QPCR | Stratagene | RRID:SCR_016375 | version 4.1 |
| Software, algorithm | ImageJ | DOI: 10.1038/nmeth.2089 | RRID:SCR_003070 | version 1.50a |
| Software, algorithm | EnVision | Perkin Elmer | RRID:SCR_016681 | version 1.12 |
| Software, algorithm | R Project for statistical computing | https://www.r-project.org | RRID:SCR_001905 | version 3.5.1 |

As described in the Registered Report (*Li et al., 2015*), we attempted a replication of the experiments reported in Figures 1B, 4A and D and Supplemental Figures 4A,B and 5C of *Liu et al. (2011)*. A detailed description of all protocols can be found in the Registered Report (*Li et al., 2015*) and are described below with additional information not listed in the Registered Report, but needed during experimentation.

## Cell culture

LAPC4 (RRID:CVCL_4744) (shared by Dr. Dean G. Tang, MD Anderson Cancer Center and Dr. Charles L. Sawyers, Memorial Sloan Kettering Cancer Center), DU145 cells (ATCC, cat# HTB-81, RRID:CVCL_0105), HT-1080 (ATCC, cat# CCL-121, RRID:CVCL_0317), and HEK293T cells (ATCC, cat# CRL-3216, RRID:CVCL_0063) were maintained in growth medium as described in the Registered Report (*Li et al., 2015*) with IMDM (Gibco, cat# 21056–023) for LAPC4 cells sourced differently than listed. LAPC4 cells from Sawyers lab, which were used in the experiments reported here, were maintained for five passages in culture prior to mouse injection. Quality control data for cell lines are available at https://osf.io/6u42f/. This includes results confirming the cell lines were free of mycoplasma contamination as well as STR DNA profiling of the cell lines. STR profile performed by (DDC Medical, Fairfield, Ohio) or the following methods: DNA from cells were extracted with a genomic DNA extraction kit (Axygen Scientific, Inc, Union City, California), amplified using the 20 - STR amplification approach, then detected and analyzed for STR loci and the sex determination gene Amelogenin using a 3730XL DNA analyzer (Applied Biosystems, Inc, Foster City, California) and GeneMapper software (RRID:SCR_014290), version 4.0. Cells were confirmed to be the indicated cell lines when queried against STR profile databases (DSMZ and Cellosaurus (*Bairoch, 2018*)).

## Animals

All animal procedures were approved by the Crown Biosciences, Inc animal use committee (Animal Use Protocol# AN-1507-007-108) and were in accordance with the Crown Biosciences policies on the care, welfare, and treatment of laboratory animals. Randomization and blinding occurred during lentiviral infection of LAPC4 cells and tumor implantation experiments. Randomization was performed based on 'matched distribution' randomization method using the multi-task method (Studylog Systems software; RRID:SCR_016682, version 3.1.399.19) on Day 0.

Six to seven week old male NOD/SCID mice (Vital River, Strain code: 406, RRID:IMSR_CRL:394) were housed in sterile conditions under standard temperature, humidity, and timed lighting conditions with 12 hr light/dark cycles, and were provided with sterile rodent chow and water *ad libitum*. Animals were monitored daily and body weight and tumor volume was recorded weekly after injection.

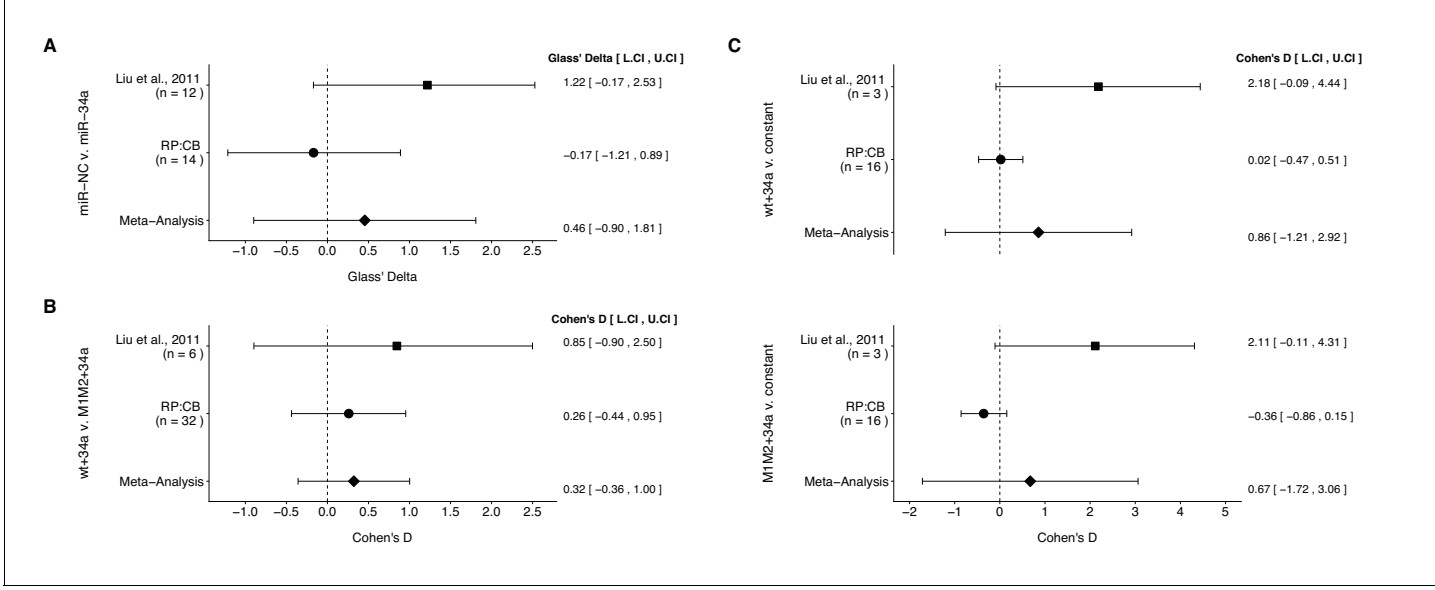

**Figure 4.** Meta-analyses of each effect. Effect size and 95% confidence interval are presented for *Liu et al. (2011)*, this replication study (RP:CB), and a random effects meta-analysis of those two effects. Cohen's *d* and Glass' delta are standardized differences between the two indicated measurements with the calculated effects for the original study effects reported as positive values. Sample sizes used in *Liu et al. (2011)* and RP:CB are reported under the study name. (A) Tumor weights from LAPC4 cells transduced to express miR-34a or negative control (meta-analysis *p*=0.510). (B) Luciferase values from cells transfected with miR-34a and wild-type luciferase reporter (wt) compared to cells transfected with miR-34a and double mutant luciferase reporter (M1M2) (meta-analysis *p*=0.353). (C) Luciferase values from cells transfected with miR-34a and wt compared to a constant of 1 (miR-NC and wt) (meta-analysis *p*=0.416), and luciferase values from cells transfected with miR-34a and M1M2 compared to a constant of 1 (miR-NC and M1M2) (meta-analysis *p*=0.581). Additional details for these meta-analyses can be found at https://osf.io/p9xn8/.
DOI: https://doi.org/10.7554/eLife.43511.008

## Xenograft tumor maintenance

After one week of acclimation, male NOD/SCID mice were implanted with a testosterone pellet as described in the Registered Report (*Li et al., 2015*). Pellets were prepared as described previously (*Nicholson et al., 2013*) with testosterone propionate (CAS: 57-85-2; Aladdin, Shanghai, China; lot# L1401028) and a 2 mm by 4 mm die set (International Crystal Laboratories, Garfield, New Jersey). Immediately after testosterone pellet implantation, mice were inoculated subcutaneously in the flank with $2 \times 10^6$ LAPC4 cells suspended in 30 µl IMDM supplemented with 10% FBS and 30 µl high concentration Matrigel (BD Biosciences, cat# 354248).

## Purification of LAPC4 cells from xenograft tumors

Tumors were excised when they reached 1 to 1.5 cm$^3$ (122–125 days after injection), weighed, and dissociated for purification of LAPC4 cells as described in the Registered Report (*Li et al., 2015*). Tumor weight and cell viability counts before and after Histopaque-1077 gradient and depletion of lineage-positive (i.e. mouse) cells using the MACS Lineage Cell Depletion Kit (Miltenyi Biotech, cat# 130-090-858) are available at https://osf.io/4jbhf/. Purified LAPC4 cells were then used for experiments.

## Lentiviral infection of LAPC4 cells and tumor implantation

Lentiviral particles were generated in HEK293T cells with psPAX2 (Crown Biosciences Inc, RRID: Addgene_12260), pMD2.G (Crown Biosciences Inc, RRID:Addgene_12259), and pMDLg/pRRE (Wuhan Miaoling Bioscience and Technology Co., cat# P0685, RRID:Addgene_12251) plasmids and viral titer was determined using HT1080 cells as described in the Registered Report (*Li et al., 2015*). Plasmids were purified using an endo-free maxiprep kit (Qiagen, cat# 12362) and identity of vectors were confirmed by sequencing and agarose gel electrophoresis (https://osf.io/rk42q/). Purified LAPC4 cells (pooled from three tumors) were transduced with lentiviral particles engineered to express pre-mir-34a (System Biosciences, cat# PMIRH34aPA-1) or scramble non-targeting pre-

**Table 1.** STR profiles.

LAPC4 cells provided by authors of the original study (*Liu et al., 2011*) and authors who originally isolated the LAPC4 cell line (*Klein et al., 1997*) underwent STR analysis for the indicated markers. The STR profiles for LAPC4 cells from databases are also provided for comparison (ATCC; *van Bokhoven et al., 2003*; DSMZ).

| Marker | Liu | Klein | ATCC | Van bokhoven | DSMZ |
|---|---|---|---|---|---|
| Amelogenin | X | X,Y | X,Y | X,Y | X,Y |
| CSF1PO | 11,12 | 12,13 | 12,13 | 12,13 | 12,13 |
| D13S317 | 10,12 | 10,11,12 | 9,10,11,12 | 10,12 | 9,10,11,12 |
| D16S539 | 9 | 9 | 9 | 9 | 9 |
| D5S818 | 11,13 | 11,13 | 12 | 11,13 | 12 |
| D7S820 | 10.3,11 | 11,12 | 10,10.3,11 | 11 | 10,10.3,11 |
| TH01 | 6,9.3 | 6,9.3 | 6,9.3 | 6,9.3 | 6,9.3 |
| TPOX | 8 | 8,9 | 8 | 8 | 8 |
| vWA | 16 | 16,17,18 | 15,16 | 16 | 15,16 |
| D18S51 | 14,21,22 | 15,21,22 | 14,15,22 | 14,15,22 | |
| D21S11 | 28,30 | 28,29,30,31 | 28,30 | 28,30 | |
| D3S1358 | 15,16,17 | 15,17,18 | 15,17 | 15,17 | |
| D8S1179 | 13 | 13,14 | 13 | 13 | |
| FGA | 21,22,23,24 | 21,22 | 21,22 | 21,22 | |
| D19S433 | 13,14,16 | 12,14,15 | | | |
| D2S1338 | 17,18,19 | 17,18,19 | | | |
| Penta D | 10 | 9 | | | |
| Penta E | 12,13 | 12,13 | | | |
| D12S391 | | 20 | | | |
| D6S1043 | | 12,13 | | | |

DOI: https://doi.org/10.7554/eLife.43511.004

miRNA (System Biosciences, cat# PMIRH000PA-1) at an MOI of 10. The lentiviral vectors use the CMV promoter to drive pre-miRNA expression and the copGFP marker under the control of the EF-1 alpha promoter. During infection LAPC4 cells were kept in a single cell suspension in ultra-low attachment flasks in growth medium at 37°C in a humidified atmosphere at 5% $CO_2$. 48 hr post-infection male NOD/SCID mice were inoculated subcutaneously in the flank with $1 \times 10^4$ LAPC4 cells suspended in 20 µl IMDM supplemented with 15% FBS and 20 µl high concentration Matrigel. Mice were randomized by body weight at time of injection and was consistent with even distribution across all groups (one-way ANOVA: $F(2,18) = 0.0003$, $p=0.9997$). Infected LAPC4 cells were confirmed to be at least 90% transduction efficiency (determined by GFP) at time of injection (representative microscopy images available at https://osf.io/hrbd8/) with aliquots of cells saved for qPCR analysis. Mice were monitored until the negative control transduced group reached an average estimated tumor weight (based on tumor volume calculations) of 0.54 grams. Mice were euthanized and primary tumors were weighed (scale: Mettler Toledo, Columbus, Ohio), with pictures of the tumors taken with a ruler for scale (available at https://osf.io/k28ab/). Tumors were divided into two pieces: one for protein extraction (snap frozen) and one for RNA extraction (immerged in RNAlater (Invitrogen, cat# AM7021) overnight and then frozen).

## Fluorescence-activated cell sorting

$2.1 \times 10^6$ purified LAPC4 cells (pooled from three tumors) were resuspended in 100 µl cold staining buffer and separated into two tubes. 20 µl FcR blocking reagent and 10 µl of FITC-conjugated monoclonal anti-CD44 (BD Biosciences, cat# 555478, clone G44-26, RRID:AB_395870) or FITC-conjugated monoclonal mouse IgG2$_b$, κ isotype control (BD Biosciences, cat# 555742, clone 27–35, RRID:

AB_396085) was added and incubated at 4 ℃ in the dark for 15 min. Flow cytometry analysis was performed on a FACSAria II (BD Biosciences) and analyzed with FACSDiva software (BD Biosciences, RRID:SCR_001456), version 6.1.3. Gating strategy is depicted in *Figure 1—figure supplement 1A*. An initial attempt was performed, which revealed few purified LAPC4 cells were CD44 positive relative to isotype control. As such, the gating strategy differed to include only the top 1% of the population, instead of the top 10%, with the remaining population constituting CD44⁻ cells. Human peripheral blood mononuclear cells (PBMC) were also tested to confirm specificity of the antibodies and staining procedure (*Figure 1—figure supplement 1B*).

## Quantitative PCR

Small RNA-containing total RNA was isolated from FACS sorted LAPC4 cells using the *mir*Vana miRNA Isolation Kit (Invitrogen, cat# AM1561) or infected purified LAPC4 cells or tumors using the *mir*Vana PARIS RNA Isolation Kit (Invitrogen, cat# AM1556) according to manufacturer's instructions. miRNA was reverse transcribed to cDNA using TaqMan MicroRNa Reverse Transcription Kit (Applied Biosystems, cat# 4366596) according to manufacturer's instructions. DNA was quantified using a Nanodrop spectrophotometer (Thermo Fisher Scientific, cat# ND-2000). qPCR was performed in technical triplicate using an MX3005P Real-time PCR System (Stratagene) with assays specific for each miRNA of interest as outlined in the Registered Report (*Li et al., 2015*). To detect copGFP expression, the following primers were used (Forward: 5'-AGGACAGCGTGATCTTCACC-3'; Reverse: 5'-CTTGAAGTGCATGTGGCTGT-3') (*Wang et al., 2008*). PCR reactions were performed in technical triplicate with the following cycling conditions [1 cycle 95℃ for 10 min – 40 cycles 95℃ for 15 s, 60℃ for 60 s] using an Mx3005P Real-time PCR System (Stratagene) and MxPro QPCR software (RRID:SCR_016375), version 4.1. Negative controls containing no cDNA template were included. Relative expression levels were determined using the ΔΔCt method.

## Western blots

Cell lysate was generated from tumor samples as described in the Registered Report (*Li et al., 2015*). Membranes were loaded with 40 µg of total protein and probed with: rabbit anti-CD44 (AbCam, cat# ab51037, clone EPR1013Y, RRID:AB_868936), 1:1000 dilution; mouse anti-ß-actin (Cell Signaling Technology, cat# 3700, clone 8H10D10, RRID:AB_2242334), 1:1000 dilution and the appropriate secondary antibody: HRP-conjugated goat anti-rabbit (Cell Signaling Technology, cat# 7074, RRID:AB_2099233), 1:2000 dilution; HRP-conjugated horse anti-mouse (Cell Signaling Technology, cat# 7076, RRID:AB_330924), 1:5000 dilution. Membranes were washed with TBST and incubated with SuperSignal West Pico Chemiluminescent Substrate (Thermo Fisher Scientific, cat# 34080) according to the manufacturer's instructions. Scanned Western blots were quantified using ImageJ software (RRID:SCR_003070), version 1.50a.

## Luciferase assay

$3 \times 10^4$ DU145 cells in 500 µl growth medium were plated in 24 well plates and incubated overnight at 37℃ in a humidified atmosphere at 5% $CO_2$. The following day, cells were checked to make sure they were 70–90% confluent at the time of transfection and new growth medium not containing penicillin-streptomycin was added. Cells were transfected with 50 µl transfection complex such that each well would contain 1 µg luciferase reporter (pMIR-CD44-3'UTR or pMIR-CD44-M1M2-3'UTR (shared by Dr. Dean G. Tang, MD Anderson Cancer Center)), 1 ng phRL-CMV (shared by Dr. Dean G. Tang, MD Anderson Cancer Center), and 24 pmoles *mir*Vana miRNA mimics (has-miR-34a-5p (Thermo Fisher Scientific, Assay ID# MC11030) or Negative Control #1 (Thermo Fisher Scientific, cat# 4464058)). Plasmids were purified using an endo-free maxiprep kit and identity of vectors were confirmed by sequencing and agarose gel electrophoresis (https://osf.io/rk42q/). 6 hr after transfection cells were replaced with fresh growth medium to increase cell viability as determined necessary during pilot assays. Cells were incubated at 37℃ in a humidified atmosphere at 5% $CO_2$. 48 hr later the luciferase and renilla signals were determined using the dual luciferase assay (Promega, cat# E1960), EnVision 2104 Multilabel Plate Reader (Perkin Elmer), and EnVision software (Perkin Elmer; RRID: SCR_016681), version 1.12, according to manufacturer's instructions. Additional method details and raw data are available at https://osf.io/vzb82/.

## Statistical analysis

Statistical analysis was performed with R software (RRID:SCR_001905), version 3.5.1 (*Core Team, 2018*). All data, csv files, and analysis scripts are available on the OSF (https://osf.io/gb7sr/). Confirmatory statistical analysis was pre-registered (https://osf.io/jhtae/) before the experimental work began as outlined in the Registered Report (*Li et al., 2015*). Data were checked to ensure assumptions of statistical tests were met and in the case of the tumor volume data were natural log transformed to achieve a normal distribution. The exploratory survival analysis was analyzed with a COX proportional hazard (CPH) model because ties, which were prevalent in the data, were better handled with CPH than a log-rank (Mantel-Cox) test. When described in the results, the Bonferroni correction, to account for multiple testings, was applied to the alpha error or the *p*-value. The Bonferroni corrected value was determined by divided the uncorrected value (0.05) by the number of tests performed. The asymmetric confidence intervals for the Cliff's *d* estimates were determined using the normal deviate corresponding to the $(1 - alpha/2)^{th}$ percentile of the normal distribution (*Cliff, 1993*). A meta-analysis of a common original and replication effect size was performed with a random effects model and the *metafor* R package (*Viechtbauer, 2010*) (https://osf.io/p9xn8/). The original study data pertaining to Figures 1B and 4D, and Supplemental Figure 5C were shared by the original authors or published in the original paper (*Liu et al., 2011*). The summary data was published in the Registered Report (*Li et al., 2015*) and used in the power calculations to determine the sample size for this study.

## Data availability

Additional detailed experimental notes, data, and analysis are available on OSF (RRID:SCR_003238) (https://osf.io/gb7sr/; *Yan et al., 2018*). This includes the R Markdown file (https://osf.io/e89mt/) that was used to compose this manuscript, which is a reproducible document linking the results in the article directly to the data and code that produced them (*Hartgerink, 2017*). Flow cytometry data for this study has also been deposited at Flow Repository (RRID:SCR_013779; *Spidlen et al., 2012*), where it is directly accessible at https://flowrepository.org/id/FR-FCM-ZYNB.

## Deviations from registered report

We obtained LAPC4 cells from the original authors, but as described above changed the source of these cells. We also modified the gating strategy when determining $CD44^+$ and $CD44^-$ populations as described above. Importantly, we included a control (PBMC) to confirm specificity of the antibodies and staining procedure (*Figure 1—figure supplement 1B*). Similarly, instead of obtaining enough purified $CD44^+$ LAPC4 cells required using multiple tumors, which were pooled, instead of individual tumors as specified in the Registered Report. When we infected purified LAPC4 cells to express miR-34a we changed the source of the lentiviral packaging plasmids and modified the time from infection to injection from 24 hr to 48 hr. The reason is because we were unable to achieve high transduction efficiency (>90% GFP positive) when the protocol was followed as described in the Registered Report. Importantly, the transduced LAPC4 cells successfully expressed miR-34a (*Figure 2A*). Additionally, because we were unable to detect miR-34a in the tumors derived from miR-34a transduced LAPC4 cells (*Figure 2—figure supplement 1C*) we analyzed the tumors to see if GFP was detectable, as an indicator of whether the tumors were derived from infected cells. Finally, we used ECL and film for the Western blots, instead of the Odyssey system outlined i the Registered Report. Additional materials and instrumentation not listed in the Registered Report, but needed during experimentation are also listed.

## Acknowledgements

The Reproducibility Project: Cancer Biology would like to thank Dr. Can Liu and Dr. Dean G Tang (MD Anderson Cancer Center) for sharing critical information, data, and reagents, specifically plasmids and LAPC4 cells used in the original study. We want to thank Dr. Charles L Sawyers (Memorial Sloan Kettering Cancer Center) for sharing LAPC4 cells. We would also like to thank Courtney Soderberg (Center for Open Science) for assistance with statistical analyses, Young Kim (System Biosciences) for helpful discussions about how to detect copGFP, and the following companies for generously donating reagents to the Reproducibility Project: Cancer Biology; American Type and

Tissue Collection (ATCC), Applied Biological Materials, BioLegend, Charles River Laboratories, Corning Incorporated, DDC Medical, EMD Millipore, Harlan Laboratories, LI-COR Biosciences, Mirus Bio, Novus Biologicals, Sigma-Aldrich, and System Biosciences (SBI).

## Additional information

### Group author details

**Reproducibility Project: Cancer Biology**
**Elizabeth Iorns**: Science Exchange, Palo Alto, United States; **Rachel Tsui**: Science Exchange, Palo Alto, United States; **Alexandria Denis**: Center for Open Science, Charlottesville, United States; **Nicole Perfito**: Science Exchange, Palo Alto, United States; **Timothy M Errington**: Center for Open Science, Charlottesville, United States

### Competing interests
Xuefei Yan, Beibei Tang, Biao Chen, Huajun Yang: Crown Biosciences Inc is a Science Exchange associated lab. Reproducibility Project: Cancer Biology: EI, RT, NP: Employed by and hold shares in Science Exchange Inc.The other authors declare that no competing interests exist.

### Funding
The Reproducibility Project: Cancer Biology is funded by the Laura and John Arnold Foundation, provided to the Center for Open Science in collaboration with Science Exchange. The funder had no role in study design, data collection and interpretation, or the decision to submit the work for publication.

### Author contributions
Xuefei Yan, Beibei Tang, Acquisition of data, Analysis and interpretation of data, Drafting or revising the article; Biao Chen, Yongli Shan, Reproducibility Project: Cancer Biology, Huajun Yang, Analysis and interpretation of data, Drafting or revising the article

### Author ORCIDs
Alexandria Denis http://orcid.org/0000-0002-1210-2309
Timothy M Errington http://orcid.org/0000-0002-4959-5143

### Ethics
Animal experimentation: All animal procedures were approved by the Crown Biosciences, Inc. animal use committee (Animal Use Protocol# AN-1507-007-108) and were in accordance with the Crown Biosciences policies on the care, welfare, and treatment of laboratory animals.

### Decision letter and Author response
Decision letter https://doi.org/10.7554/eLife.43511.017
Author response https://doi.org/10.7554/eLife.43511.018

## Additional files

### Supplementary files
• Transparent reporting form
DOI: https://doi.org/10.7554/eLife.43511.009

• Reporting standard 1. The ARRIVE guidelines checklist.
DOI: https://doi.org/10.7554/eLife.43511.010

### Data availability
Additional detailed experimental notes, data, and analysis are available on OSF (RRID:SCR_003238) (https://osf.io/gb7sr/; Yan et al., 2018). This includes the R Markdown file (https://osf.io/e89mt/) that

was used to compose this manuscript, which is a reproducible document linking the results in the article directly to the data and code that produced them (Hartgerink, 2017). Flow cytometry data for this study has also been deposited at Flow Repository (RRID:SCR_013779; Spidlen et al., 2012), where it is directly accessible at https://flowrepository.org/id/FR-FCM-ZYNB.

The following datasets were generated:

| Author(s) | Year | Dataset title | Dataset URL | Database and Identifier |
|---|---|---|---|---|
| Yan X, Tang B, Chen B, Shan Y | 2018 | Study 28: Replication of Liu et al., 2011 (Nature Medicine) | http://dx.doi.org/10.17605/OSF.IO/GB7SR | Open Science Framework , 10.17605/OSF.IO/GB7SR |
| Yan X, Tang B, Chen B | 2018 | Replication Study: The microRNA miR-34a inhibits prostate cancer stem cells and metastasis by directly repressing CD44 | https://flowrepository.org/id/FR-FCM-ZYNB | Flowrepository, FR-FCM-ZYNB |

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
