## [Decision Letter]

Thank you for submitting your article "Replication Study: The microRNA miR-34a inhibits prostate cancer stem cells and metastasis by directly repressing CD44" for consideration by *eLife*. Your article has been reviewed by four peer reviewers, and the evaluation has been overseen by a Reviewing Editor and Sean Morrison as the Senior Editor. Andrea Ventura (Reviewer #1) and Julie Liu (Reviewer #4) have agreed to reveal their names.

Summary:

Your manuscript is recommended for publication in *eLife* pending satisfactory revisions of several remaining points. Specifically:

1) Could the replication authors confirm whether their departure from the original conditions used by Liu et al. was approved as part of the original registered protocol? Could the authors elaborate and discuss the path to making this decision and the potential confounds?

2) The observation that in the miR-34 over-expression experiment, miR-34 over-expression is lost raises concerns, as described by reviewer #2. The authors need to address this concern.

3) Address as much as possible the minor points identified by Referees #1 and #3.

*Reviewer #1:*

This manuscript by Yan and colleagues is part of the "Reproducibility Project" and was aimed at replicating key experiments from the paper: "The microRNA miR-34a inhibits prostate cancer stem cells and metastasis by directly repressing CD44".

The original paper provided experimental evidence showing that miR-34a expression is reduced in CD44+ cells compared to CD44- cells prostate cancer cells and that ectopic expression of miR-34a is sufficient to strongly repress tumor formation in a xenograft assay. Moreover, the authors provided data showing that miR-34a exerts this potent tumor repressive activity at least in part by targeting the cell adhesion molecule CD44.

In the replication study, the Yan et al. fail to replicate key portions of the original study. In particular, they fail to detect significant down-regulation of miR-34a in CD44-Hi LPCA4 cells, fail to observe tumor growth inhibition when LAPC4 cells transduced with a miR-34a expressing lentivirus were injected in immunocompromised mice, and fail to detect significant repression by ectopic miR-34a in luciferase reporter assays. These results call into question the central message of the original publication.

The manuscript is well written, the experiments are clearly illustrated and their results are generally consistent with the authors' interpretation. The relevant scientific literature is appropriately cited and the statistical analysis performed to reject the various null hypothesis is, as far as I can tell, sound.

The authors have also to be commended for carefully listing potential explanations for the discrepancy between their results and those reported in the original manuscript.

In particular, the fact that the LAPC4 cell line initially obtained by Liu and colleagues failed to generate tumors and had to be replaced by LAPC4 from a different source is worrisome, given the high genetic drift of cancer cell lines.

Equally perplexing is the observation that expression of the miR-34a transgene appears silenced in tumors isolated from mice injected with LAPC4 cells transduced with the mIR-34a overexpressing. This may suggest that there is a strong selection against miR-34a in vivo, but it doesn't explain why the same effect was not observed by Liu and colleagues. It is important to point out that there is abundant experimental evidence that ectopic expression of miR-34a can result in cell cycle arrest and apoptosis, although the evidence that loss of endogenous miR-34a is sufficient to promote or accelerate tumorigenesis is less conclusive.

Overall, this is a strong manuscript that, in my opinion deserves publication.

Minor issues:

The authors seem to imply that reduced expression of the luciferase reporter carrying the 3'UTR of CD44 with point mutations disrupting the miR-34a sites could be explained by the fact that the WT 3'UTR contains a potential miR-141 binding site overlapping one of the two miR-34a site. Unless there is something I have missed, I find this explanation illogical. If the putative miR-141 site is also disrupted in the mutant 3'UTR one would expect increased luciferase activity, not reduced (loss of repression by miR-141). The authors should either remove or clarify this sentence.

"To summarize, we found results that were in the same direction as the original study and not statistically significant where predicted". This sentence is unclear.

*Reviewer #2:*

It appears that multiple conclusions drawn from the original manuscript by Liu et al., are not confirmed by this replication study. It is unfortunate that based on the flow analysis, the replication authors were forced to deviate from the original study's experimental protocol when choosing the cut-off for CD44 high or CD44 low. Two deviations were made. The first was to use a 1% cutoff rather than the prior manuscripts' 10% cutoff. The second was to assign the other 99% of the population to a CD44 low status. The authors could argue that 10% of the replication study's CD44 low population (9%/99%) is in fact CD44 high. I understand the replication authors' potential reason for doing this: that there wasn't a clear CD44 high versus CD44 low cutoff. However, it does represent a significant departure from the prior work. Could the replication authors confirm whether this departure was approved as part of the original registered protocol? Could the authors perhaps elaborate and discuss the path to making this decision and the potential confounds?

The second concern is the observation that in the miR-34 over-expression experiment, miR-34 over-expression is lost. One hand, one may conclude that miR-34 over-expression in this experiment did not reduce tumor growth, given the data in the figure. However, there is a hazard in this, since the replication authors have themselves shown that miR-34 expression is silenced. Such silencing typically occurs when the microRNA is indeed suppressive to cancer progression (the original authors' conclusion). One could imagine that such silencing may have imparted a benefit at an early initiation step (tumor re-initiation) and been maintained. Thus, this experimental observation does not allow one to draw the conclusion that miR-34 *does not* suppress tumor growth or progression in this system, since miR-34 over-expression is not confirmed in vivo.

The luciferase reporter experiment clearly shows that exogenously transfected miR-34a does not repress the wildtype reporter signal.

*Reviewer #3:*

As a part of "Reproducibility Project: Cancer Biology, this manuscript aims to duplicate the results of a previous publication "The microRNA miR-34a inhibits prostate cancer stem cells and metastasis by 41 directly repressing CD44" (Liu et al., 2011). The authors generated different observations from those reported by the Tang group, using the same/similar reagents and protocols whenever possible. The original paper by Liu et al. claimed that miR-34a negatively regulate CD44 in prostate cancer stem cells, as they reported the downregulation of miR-34a in CD44 high LAPC4 prostate cancer cells (the putative stem cell population), the suppression of tumor development and Cd44 downregulation by miR-34a overexpression in xenograft models, and the miR-34a dependent regulation of Cd44 in luciferase assays. However, the replication study made different observations, including a 2-fold increase of miR-34a in CD44 high versus CD44 low LAPC4 prostate cancer cells, no effect on tumor growth or Cd44 expression by miR-34a overexpression in the xenograft model, and no miR-34a-dependent regulation on CD44.

This manuscript has raised questions about the reproducibility of the original report by Liu et al. This reviewer agrees with the authors that the selected experiments constitute the core discoveries reported by the original paper, and should be the focus of reproducibility effort. Of particular importance is the biological effects of miR-34 in prostate cancer development, and the miR-34a dependent regulation of CD44 in prostate cancer cells. The authors have carefully designed and executed their experiments, they included appropriate controls wherever possible, and obtained solid results based on the reported experimental conditions. It is clear to this reviewer that these authors cannot reproduce some of the most important results from the original report. This paper is suited for publication at *eLife* upon revision.

Major comments:

1) The key issue encountered by the reproducibility study is the cell line of choice. The LAPC4 cell line acquired from the Tang group only exhibited a partial match with the standard LAPC4 profiles in the DSMZ and Cellosaurus database, and more importantly, failed to generate xenograft tumors in NOD/SCID mice. Hence the authors had to acquire cytologically and functionally validated LAPC4 cells from a different source to perform this reproducibility study. This reviewer is wondering if the authors have contacted the Tang group to acquire additional information about the LAPC4 cell lines they had problems with, and made a second request for LAPC4 from the Tang group. It is unclear if the failure to reproduce the original observations is largely due to the cell line difference, or if there exist other contributing factors.

2) To validate the tumor suppressor functions of miR-34a, the authors generated LAPC4 cells that overexpress miR-34a by lentiviral transduction, but observed no effect on tumor development or CD44 expression in xenograft models. However, miR-34a expression is silenced in vivo specifically in LAPC4 cells transduced by mir-34a lentiviral vectors, but not in control transduced or non-transduced LAPC4 cells. This observation implies that miR-34a expression could have a tumor suppressor function, hence its expression is selected against in vivo (this is consistent with the tumor suppressor effects of miR-34a in overexpression studies in other models). Again, the cell line difference could contribute significantly to the different observations between the two studies. It would be nice to have the authors provide a more in-depth discussion about this discrepancy in results.

3) The original study reported a direct regulation between miR-34a and CD44 3'UTR, but the extent of regulation reflected in luciferase assays was rather modest. Luciferase assays have been widely used to validate a direct regulation of a target mRNA by a miRNA, yet in my opinion, these assays are not very reliable, particularly for miRNA targets that are modestly regulated. Often times, these assays rely on overexpression of both miRNAs and luciferase reporters with wildtype or mutated 3'UTRs to determine the extent of their direct interactions in cell lines. In addition to the binding between miRNAs and the target 3'UTR, the relative expression levels between miRNAs and reporter, which often deviates from physiological conditions, play an important role in the results of the assay. Hence, I suggest that the authors include such discussion in their revised manuscript.

Minor comments:

1) It is not appropriate to claim that "miR-34a was overexpressed in CD44+ cells, being expressed at 202% of the level in the CD44- 139 cells 140 (Figure 1)". miR-34a is elevated in level in CD44+ cells, but not necessarily overexpressed. Please be precise when describing this observation.

*Reviewer #4:*

I strongly oppose the publication of this 'reproducibility' manuscript, for the following considerations.

First of all, the conclusion of the original paper that miR-34a inhibits prostate cancer stem cells and metastasis by directly repressing CD44 (Liu et al., 2011) was based on experimental results obtained in multiple prostate cancer cell line and xenograft models as well as human primary prostate cancer cells. Each experiment was scientifically designed and rigorously performed with biological and technical repeats. That paper has been highly cited (more than 1150 times) since the publication. Besides that, tumor suppressive functions of miR-34a have also been intensively studied and corroborated in many tumor systems by independent researchers. Since the publication of the original paper, there have been 32 publications that are directly or indirectly in support of the original findings regarding miR-34a and CD44. However, in this Reproducibility Project, the authors are trying to use shoddy data from poorly designed and executed experiments to discredit the original paper. Two of the three experiments in this manuscript were done in only one xenograft model LAPC4. All of the three experiments were one-time experiments with limited number of sample size, which lacks proper biological and technical repeats, especially when they saw big variations within the experimental groups. These points challenge the credibility of the authors' results, which were obtained via experiments that lacked the basic scientific rigor and could not be used to draw any meaningful conclusions.

Secondly, the LAPC4 cells from the original paper have been validated by cell line authentication in our institute (MD Anderson Cancer Center), and have been intensively used in our lab for more than 12 years. We trust our cells and we are more than willing to provide our LAPC4 cells to them for the in vivo studies.

Thirdly, the authors mentioned that they did not see tumor suppressive function of miR-34a in the LAPC4 model; however, they also mentioned that miR-34a over-expression in the LAPC4 tumor cells was already gone at the endpoint of the tumor experiments. That perfectly explains why they didn't see a tumor suppressive function of miR-34a. Using this kind of data to claim that miR-34a lacks tumor suppressive function is merely unacceptable.

Lastly, in the luciferase reporter experiment, there is huge variation in luciferase activity even within one experimental group. It's highly unscientific and irresponsible to make any conclusion from this kind of data.

Collectively, publication of this shoddily done 'Reproducibility' manuscript that lacks the basic scientific rigor will only generate confusions in the field and do a big disservice and damage to the entire scientific community.

---

## [Author Response]

Thank you for the evaluation of our manuscript. We would like to thank the reviewers for their attention and insightful commentary. We appreciate their feedback in strengthening this manuscript and welcome the opportunity to publish with *eLife*.

1) Could the replication authors confirm whether their departure from the original conditions used by Liu et al. was approved as part of the original registered protocol? Could the authors elaborate and discuss the path to making this decision and the potential confounds?

The cut-off of CD44 high or CD44 low cells described in the Registered Report, which was informed from the original paper and original authors, was approved at 10%. The departure was not approved as part of the Registered Report and occurred during the course of experimentation. In the revised manuscript we more explicitly state that the deviation was from not only the original paper, but the Registered Report. We also elaborated on the process, specifically the expectation that the cellular population would display CD44 staining along a spectrum that was clearly distinguishable from controls. Instead we found little to no separation of CD44 positive cells from isotype control. Thus, the replication attempt focused on measuring the expression of the target microRNAs in CD44 high (i.e. positive) cells vs CD44 low (i.e. negative) cells. Additionally, considering the hypothesis being tested was reduced miR-34a levels in CD44^+^ cells, the deviation from top 10% to 1% was considered a way to prevent the CD44^-^ population, which was substantially larger, based on staining controls, from masking the reduced levels of the microRNAs in the few CD44^+^ cells observed. That is, if we took the top 10% it would contain 90% CD44^-^ cells and 10% CD44^+^ cells, essentially masking any attempt to observe if there was reduced expression of miR-34a in CD44^+^ cells. While this was a substantial deviation that could mask differences in expression, it is important to note that we observed let-7b levels were reduced in CD44^+^ cells, relative to CD44^-^ cells, as expected and similar to the original study. However, we understand reviewer #2’s concern that this creates a potential confound, particularly in the CD44 low population that could contain CD44 high cells and have included the potential confounds in the revised manuscript.

2) The observation that in the miR-34 over-expression experiment, miR-34 over-expression is lost raises concerns, as described by reviewer #2. The authors need to address this concern.

We agree with the comments that the loss of miR-34a over-expression, although not through failure of maintaining the sequence as determined by copGFP expression, complicates the interpretation of the experiment. We have incorporated additional discussion in the revised manuscript on the implications of what can be drawn from these results as well as other possible factors (i.e. cell line differences) and mechanisms (i.e. benefit at an early initiation step) that could have influenced this result.

3) Address as much as possible the minor points identified by Referees #1 and #3.

We addressed the minor points by Referees #1 and #3, with responses listed below.

Reviewer #1 Minor issues:

The authors seem to imply that reduced expression of the luciferase reporter carrying the 3'UTR of CD44 with point mutations disrupting the miR-34a sites could be explained by the fact that the WT 3'UTR contains a potential miR-141 binding siteoverlapping one of the two miR-34a site. Unless there is something I have missed, I find this explanation illogical. If the putative miR-141 site is also disrupted in the mutant 3'UTR one would expect increased luciferase activity, not reduced (loss of repression by miR-141). The authors should either remove or clarify this sentence.

We removed this sentence in the revised manuscript.

"To summarize, we found results that were in the same direction as the original study and not statistically significant where predicted". This sentence is unclear.

We’ve revised this sentence to: “To summarize, we found results that were not

consistent in direction with the original study and not statistically significant.”

Additionally, this summary sentence, which is included for all Replication Studies as

suggested by eLife, summarizes two key aspects of each experiment - the direction

and statistical significance of each replication result compared to the original study.

Reviewer #3:

[…] 1) The key issue encountered by the reproducibility study is the cell line of choice. The LAPC4 cell line acquired from the Tang group only exhibited a partial match with the standard LAPC4 profiles in the DSMZ and Cellosaurus database, and more importantly, failed to generate xenograft tumors in NOD/SCID mice. Hence the authors had to acquire cytologically and functionally validated LAPC4 cells from a different source to perform this reproducibility study. This reviewer is wondering if the authors have contacted the Tang group to acquire additional information about the LAPC4 cell lines they had problems with, and made a second request for LAPC4 from the Tang group. It is unclear if the failure to reproduce the original observations is largely due to the cell line difference, or if there exist other contributing factors.

We attempted to reach out to the original authors again, but were informed that work on miR-34a and LAPC4 cells had stopped in the Tang lab and, unfortunately, they could not assist us. The potential confounds of this, specifically cell line genetic drift, have been described in the manuscript.

2) The original study reported a direct regulation between miR-34a and CD44 3'UTR, but the extent of regulation reflected in luciferase assays was rather modest. Luciferase assays have been widely used to validate a direct regulation of a target mRNA by a miRNA, yet in my opinion, these assays are not very reliable, particularly for miRNA targets that are modestly regulated. Often times, these assays rely on overexpression of both miRNAs and luciferase reporters with wildtype or mutated 3'UTRs to determine the extent of their direct interactions in cell lines. In addition to the binding between miRNAs and the target 3'UTR, the relative expression levels between miRNAs and reporter, which often deviates from physiological conditions, play an important role in the results of the assay. Hence, I suggest that the authors include such discussion in their revised manuscript.

We included a brief discussion on some important factors to consider when interpreting results from experiments that overexpress microRNAs in the revised manuscript.

Minor comments:1) It is not appropriate to claim that "miR-34a was overexpressed in CD44+ cells, being expressed at 202% of the level in the CD44- 139 cells 140 (Figure 1)". miR-34a is elevated in level in CD44+ cells, but not necessarily overexpressed. Please be precise when describing this observation.

Thank you for catching this. We have revised this sentence to appropriately describe miR-34a as being elevated, not overexpressed.